# Rate-aware Compression for NeRF-based Volumetric Video

Zhiyu Zhang*
Shanghai Jiao Tong University
Shanghai, China
zhiyu-zhang@sjtu.edu.cn

Guo Lu*
Shanghai Jiao Tong University
Shanghai, China
luguo2014@sjtu.edu.cn

Huanxiong Liang
Shanghai Jiao Tong University
Shanghai, China
huanxiong@sjtu.edu.cn

Zhengxue Cheng†
Shanghai Jiao Tong University
Shanghai, China
zxcheng@sjtu.edu.cn

Anni Tang
Shanghai Jiao Tong University
Shanghai, China
memory97@sjtu.edu.cn

Li Song†
Shanghai Jiao Tong University
Shanghai, China
song_li@sjtu.edu.cn

## Abstract

The neural radiance fields (NeRF) have advanced the development of 3D volumetric video technology, but the large data volumes they involve pose significant challenges for storage and transmission. To address these problems, the existing solutions typically compress these NeRF representations after the training stage, leading to a separation between representation training and compression. In this paper, we try to directly learn a compact NeRF representation for volumetric video in the training stage based on the proposed rate-aware compression framework. Specifically, for volumetric video, we use a simple yet effective modeling strategy to reduce temporal redundancy for the NeRF representation. Then, during the training phase, an implicit entropy model is utilized to estimate the bitrate of the NeRF representation. This entropy model is then encoded into the bitstream to assist in the decoding of the NeRF representation. This approach enables precise bitrate estimation, thereby leading to a compact NeRF representation. Furthermore, we propose an adaptive quantization strategy and learn the optimal quantization step for the NeRF representations. Finally, the NeRF representation can be optimized by using the rate-distortion trade-off. Our proposed compression framework can be used for different representations and experimental results demonstrate that our approach significantly reduces the storage size with marginal distortion and achieves state-of-the-art rate-distortion performance for volumetric video on the HumanRF and ReRF datasets. Compared to the previous state-of-the-art method TeTriRF, we achieved an approximately -80% BD-rate on the HumanRF dataset and -60% BD-rate on the ReRF dataset.

## CCS Concepts

• **Computing methodologies** → **Computer graphics**; **Computer vision**; **Virtual reality**; **Image compression**.

*Both authors contributed equally to this research.
†Corresponding authors.

## Keywords

Volumetric Video, NeRF, Compression, Rate Estimation

**ACM Reference Format:**
Zhiyu Zhang, Guo Lu, Huanxiong Liang, Zhengxue Cheng, Anni Tang, and Li Song. 2024. Rate-aware Compression for NeRF-based Volumetric Video. In *Proceedings of the 32nd ACM International Conference on Multimedia (MM '24), October 28-November 1, 2024, Melbourne, VIC, Australia*. ACM, New York, NY, USA, 10 pages. https://doi.org/10.1145/3664647.3680970

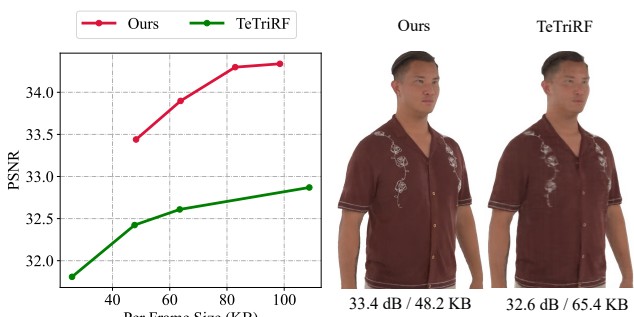

**Figure 1: Comparison of compression performance with the state-of-the-art method, TeTriRF [52]. Compared to TeTriRF, our method achieves approximately 1 dB higher PSNR at a similar bitrate, and the BD-rate is -83%.**

## 1 Introduction

3D volumetric video can provide immersive experience for viewers and exhibits a potential trend to be the next-generation video format. Previous works for volumetric videos reconstruction includes point cloud-based approaches [15] and depth-based approaches [3]. Recently, the popularized 3D representation known as Neural Radiance Fields (NeRF) [32], acclaiming for its photorealistic rendering capabilities from novel viewpoints, have attracted tremendous attention.

NeRF [32] utilizes Multilayer Perceptron (MLP) to model 3D scenes and employ volume rendering techniques to generate images from novel perspectives. To increase the training and rendering speeds, the mainstream works [6, 34, 44] incorporate explicit NeRF representations, such as grids [44], planes [6], and hash tables [34]. Furthermore, various methods [5, 10, 11, 28, 35, 36, 50]

have been proposed to extend NeRF of static scenes to dynamic scenes, i.e., volumetric videos. However, the majority of existing research primarily emphasizes enhancing the reconstruction quality of NeRF representations, while often overlooks the critical need for reducing the storage size and transmission bandwidth. This oversight presents significant challenges for practical applications, particularly in dynamic volumetric video scenarios.

To address these problems, several methods [14, 24, 25, 30, 41, 46] have been proposed to compress explicit NeRF representations for static scenes and dynamic scenes [48, 49, 52]. These methods utilize prediction, transformation, quantization, and entropy coding techniques, inspired from traditional video compression algorithms to compress the dynamic NeRF representations after the training stage, achieving significant compression rates. Among these approaches, ReRF [48] utilizes grid-based explicit representations to model dynamic scenes and devises a compression method akin to JPEG [47] to compress the representations. VideoRF [49] builds upon ReRF [48] by integrating traditional 2D video codec into the compression process. TeTriRF [52] employs a more compact tri-plane representation to model dynamic NeRF and utilizes an HEVC codec to compress the tri-plane representations. However, these methods concentrate exclusively on compressing dynamic NeRF representations post-training, neglecting the rate-distortion trade-off during the training phase. Therefore, they fail to adequately reduce spatial and temporal redundancy within the NeRF representations, thus their compression performance is far from optimal.

In this paper, we propose a rate-aware compression method tailored for NeRF-based volumetric video. Our approach estimates the bitrate of NeRF representations during its training stage and incorporates rate and distortion terms into the loss function, enabling end-to-end training. Therefore, we can obtain a compact NeRF representation with optimal rate-distortion performance. Specifically, first of all, targeting explicit NeRF grid representations, we propose inter prediction-based dynamic modeling technique by learning residual information based upon the previous frame, which effectively reduces the entropy of the NeRF representation. Second, we propose an adaptive quantization strategy with learnable quantization step to preserve more detailed information which contribute better reconstruction quality at different locations and scales. Third, in order to incorporate bitrate constraints into the training process and leverage rate-distortion loss, we introduce a tiny MLP-based implicit entropy model to estimate the rate. Given the potentially complex distribution of the explicit NeRF representation, we use temporal and spatial context for a more accurate rate estimation. Experimental results demonstrate that by applying our methodology to grid-based explicit representation [7], we can achieve state-of-the-art rate-distortion compression performance on the representative HumanRF [17] and ReRF [48] datasets. Compared with SOTA method TeTriRF [52], we achieve an approximate -80% BD-rate on HumanRF dataset and -60% BD-rate on ReRF dataset.

Our main contributions can be summarized as follows:

- We proposed a rate-aware compression framework for NeRF grid representations. Our pipeline introduces adaptive quantization strategy and spatial-temporal implicit entropy model to achieve joint rate-distortion optimization during training,

which greatly enhances the compression performance than post-training methods.

- Extensive experimental results on the benchmark datasets demonstrate the effectiveness of our approach. We can save more than 80% bitrate when compared with state-of-the-art dynamic NeRF compression approach at the same reconstruction quality.

## 2 Related Work

### 2.1 NeRF for Scene Representation

NeRF [32] enables the generation of images from arbitrary viewpoints through volume rendering, by employing a large-scale Multilayer Perceptron (MLP) to fit the colors and densities of sampled points in a scene. However, utilizing a large-scale MLP to represent the scene for real-time rendering is infeasible, prompting several works to propose combining explicit representations with smaller MLPs as a substitute for the pure implicit representation of the scene. This approach aims to reduce the computational complexity associated with large-scale MLPs. Some existing works have explored various approaches in this regard. For instance, [12] utilizes an octree, [44] employs a voxel grid, [34] implements hash tables, and [6] utilizes tensors.

Expanding NeRF [32] to dynamic scenes is not a trivial task, necessitating consideration of object movements within dynamic scenes. [9, 13, 53] model dynamic scenes by using time as an additional condition for the implicit MLP, yet these purely implicit modeling methods are not only inefficient but also perform poorly in motion-intensive settings. Alternatively, [26, 27, 35] incorporate motion modeling of the scene using a deformation field to predict displacements in dynamic scenes between each frame and a canonical frame. To expedite training and rendering, [10, 11] introduce explicit representations to accelerate the reconstruction of dynamic scenes. Essentially, these approaches utilize a single NeRF model to fit all frames within a dynamic scene, achieving commendable reconstruction results for short sequences. However, the reconstruction quality dramatically declines with longer sequences. Consequently, to ensure high-quality reconstructions in long-sequence settings, we adopt a frame-by-frame modeling approach for dynamic scenes in this paper.

### 2.2 NeRF Representations Compression

Although NeRF has achieved superior rendering quality and fast training speed via the explicit representations, it is at the cost of additional model size and storage cost. Recently, numerous works [14, 20, 24, 25, 30, 38, 41, 46, 57] have focused on compressing the explicit representation of NeRF in static scenes. [24] employs post-processing techniques to compress trained voxel grids through voxel pruning and vector quantization. [46] and [41] introduce compression-related operations during training, such as vector quantization [46] and binarization [41] to compress neural radiance fields, yet they do not consider rate-distortion optimization during training. [14, 20, 30] reduce the entropy of the NeRF representation by introducing rate loss during training, but their methods of rate estimation are relatively simplistic, and the compression performance is not optimal. NeRFCodec [25] compresses NeRF representations by adjusting encoder and decoder heads, building on

the reuse of 2D neural image codec, but this method is only applicable to plane-based NeRF representations. Moreover, these methods solely address compression in static scenes and do not account for temporal redundancy in dynamic scenes.

For dynamic NeRF compression, the recent works [16, 48, 49, 52, 56, 58] adopt frame-by-frame modeling, which enables high-quality reconstruction for long-duration sequences. These approaches also incorporate compression algorithms to ensure efficient storage utilization. ReRF [48] models the radiance field as a combination of motion fields and residual fields and employs traditional video compression pipelines to compress the representation of the radiance field. Guo *et al.* [16] adopts a vector quantization approach to eliminate the spatial-temporal redundancy of the radiance field. TeTriRF [52] directly models the radiance field as a tri-plane representation and applies a 2D video encoder to compress the representation. However, these methods do not consider the optimization of radiance field representation guided by rate-distortion loss functions. Our proposed method not only eliminates temporal redundancy but also leverages rate-distortion functions to optimize the representation of the radiance field, achieving optimal rate-distortion performance.

## 2.3 Image, Video and 3D Content Compression

To achieve efficient transmission, data compression has been investigated for other multimedia data, including image, video and 3D contents. This work also inspires from these previous approaches, thus we introduce them briefly.

In the field of image compression, various methods have been developed to formulates the rate-distortion optimization problem as an end-to-end training process to enhance compression efficiency utilizing neural networks [1, 8, 18]. Balle [1] proposed a hyperprior-based autoencoder to achieve better compression performance than traditional codecs, such as JPEG [47] and JPEG2000 [42]. Cheng [8] have employed discrete Gaussian mixture models to estimate the distribution of latents, achieving compression performance that exceeds the VVC-intra [4]. For video compression algorithms, traditional video coding standards [4, 45] utilize a hybrid coding framework that compresses video through intra prediction, inter prediction, transformation, quantization, and entropy coding. Neural network-based end-to-end video compression methods [19, 21–23, 29, 40] employ neural networks to replace modules within the hybrid coding framework. By using rate-distortion optimization to tune the entire network, the SOTA method have achieved compression performance surpassing the latest VVC standard [4].

3D content comes in various forms, such as NeRF, point clouds, and multi-view videos. For point cloud compression, MPEG has introduced G-PCC and V-PCC [39] to compress different types of point clouds. With the advancement of neural networks in image and video compression, numerous studies [37, 43] have explored neural network-based point cloud compression. [54] introduced 3D wavelet transform to compress volumetric images. [37] suggested enhancing the utilization of spatio-temporal information by leveraging voxelized information from neighboring nodes within an octree architecture, thereby further improving the efficiency of point cloud compression. For the compression of multi-view videos, MPEG has introduced MIV [3] standard, which encodes multi-view videos using 2D encoders, eliminates inter-view redundancies, and

utilizes novel view synthesis algorithms to synthesize new viewpoints. However, spatio-temporal redundancy reduction has not yet fully investigated for NeRF representations.

## 3 Preliminaries

NeRF [32] is a continuous 3D scene representation technique that learns a mapping fuction $g_\phi(\mathbf{x}, \mathbf{d}) : \mathbf{R}^d \rightarrow \mathbf{R}^c$, transforming the coordinates $\mathbf{x} = (x, y, z)$ of sampled points along a ray and the viewing direction $\mathbf{d} = (\theta, \phi)$ into color $\mathbf{c}$ and density $\sigma$:

$$(\mathbf{c}, \sigma) = g_\phi(\mathbf{x}, \mathbf{d}). \tag{1}$$

When rendering images from novel viewpoints, given a target camera extrinsic, the color $\hat{C}(\mathbf{r})$ of corresponding pixel can be obtained through volume rendering:

$$\hat{C}(\mathbf{r}) = \sum_{i=1}^{N} T_i \alpha_i \mathbf{c}_i,$$

$$T_i = \prod_{j=1}^{i-1} (1 - \alpha_i), \quad \alpha_i = 1 - \exp(-\sigma_i \delta_i). \tag{2}$$

where $T_i$ and $\alpha$ represents the transmittance and alpha value of $i$-th sampled point and $\delta_i$ denotes the distance between adjacent sampled points.

The original NeRF [32] employs a purely implicit MLP to approximate mapping function $g$. Subsequent works have introduced explicit representations (such as grids [7], planes [6], etc.) combined with a tiny rendering MLP to accelerate the training and rendering. It should be noted that our proposed compression method can be used for different explicit representations and we provide further analysis in our experimental part. Here, we adopt the DiF [7] as our default static representation in this paper and we extend it for the dynamic scenes.

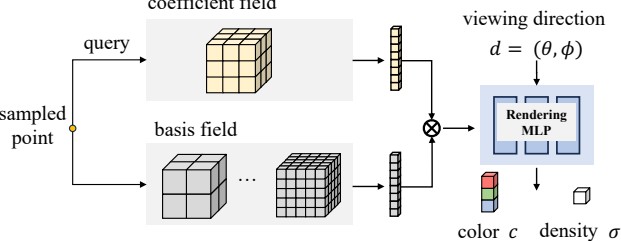

**Figure 2: Demonstration of DiF representation.**

DiF employs a grid-based explicit representation which decomposes the representation into a coefficient field and a basis field. As illustrated in Fig 2, the coefficient field comprised of a single-scale grid and the basis field consisting of six multi-scale grids. During rendering, features are initially queried from the coefficient grid and the basis grids through trilinear interpolation, followed by the fusion of features via the Hadamard product. Finally, a tiny rendering MLP maps the feature to color $\mathbf{c}$ and density $\sigma$.

**Inter Prediction-based Dynamic Modeling**

Figure 3: The training pipeline of our proposed method. (a) Firstly, the reconstructed representation $\hat{G}_{t-1}$ from the previous frame is retrieved from the decode buffer, and based on this, the residual representation $R_t$ is trained. (b) During training, adaptive quantization is utilized to enable representations at different scales to learn the optimal quantization step. Additionally, the spatio-temporal context implicit entropy model is used to estimate the bitrate of the explicit representation $R_t$. Ultimately, rate-distortion optimization is performed by integrating distortion loss and rate loss.

## 4 Methodology

Fig. 3 illustrates the training pipeline of our method. This includes a dynamic modeling method based on inter prediction, an adaptive quantization strategy, and bitrate estimation for the radiance field representation through a spatio-temporal implicit entropy model, combined with distortion loss for rate-distortion optimization.

### 4.1 Inter Prediction-Based Dynamic Modeling

Given that directly modeling entire dynamic scenes using NeRF representation like [5, 11] may result in poor performance for long sequences [48], we extend the current static NeRF representation DiF to dynamic scenes through a frame-by-frame inter prediction based modeling approach in the time dimension.

Specifically, the components required for volume rendering of a static scene include the explicit representation, occupancy grid, and the rendering MLP. These are also the components that need to be compressed and transmit. Tab 1 demonstrates the size of different components in the grid-based representation [7]. It is evident that independently modeling a NeRF for each frame without compression would require about 600 MBps (20MBx30fps) bandwidth for volumetric video, which is unacceptable. Given that the explicit representation occupies the majority of the model size, eliminating the inter-frame redundancy of the explicit representation could make dynamic NeRF more compact.

**Table 1: Model Size of different components in DiF [7].**

| Explicit Repr. | Occupancy Grid | Rendering MLP | Total |
|---|---|---|---|
| 19.21 MB | 1.08 MB | 0.17 MB | 20.46MB |

To simplify the description, we collectively refer to the coefficient grid and the basis grid as the feature grid $G$. To model dynamic

scenes, we use a simple yet effective dynamic frame-by-frame modeling strategy. Here, we first divide the entire sequence into equally long groups, where the first frame of each group is an I-frame, modeled independently, and subsequent frames are P-frames, modeled only in terms of the residual relative to the previous frame. Specifically, when modeling a P-frame, the reconstruction grid of the previous frame $\hat{G}_{t-1}$ is retrieved from the decoded buffer, and we can learn the residual grid $R_t$ for the current frame. Then the representation for current frame is expressed as:

$$G_t = \hat{G}_{t-1} + R_t. \tag{3}$$

Finally, $\hat{G}_t$ is stored into decode buffer for the reconstruction of the next frame. Furthermore, to ensure the temporal continuity and facilitate compression, we have applied an $\mathcal{L}1$ regularization to the magnitude of the residual grid, *i.e.,* $\mathcal{L}_{reg} = \|R_t\|_1$.

In addition, given the feature-level similarity of frames within the same group, we allow frames within a group to share the rendering MLP. This strategy significantly reduces bitrate consumption at a lower bitrate range without incurring substantial performance degradation.

### 4.2 Adaptive Quantization

In our proposed framework, the learned representation $G_t$ ($G_t$ for I-frame and $R_t$ for P-frame) for the current frame should be quantized before the actual entropy coding. One straightforward approach is to use a uniform quantization step for all the items in $G_t$. However, the importance of different regions/scales in the learned representation $G_t$ may vary and uniform quantization may be not the optimal solution.

In this paper, we have adopted an adaptive quantization training strategy to identify the optimal quantization step for NeRF representations. Taking DiF representation as an example, their explicit representations encompass grids at different scales. During training, we assign different quantization step parameters to grids of different

scales. These parameters are set as trainable and are continuously optimized throughout the training process. Then quantization is defined as $\hat{G}_t = \left\lfloor \frac{G_t}{q_t} \right\rceil$, where $q_i$ is trainable quantization step.

Fig. 4 illustrates the distribution histograms of quantized explicit NeRF representations obtained through adaptive quantization training versus fixed quantization training. Fig. 4 demonstrates that the representations obtained from adaptive quantization training have a wider distribution range of [-150, 150] and a larger variance. On the other hand, the representations obtained from fixed quantization training are concentrated within the range of [-20, 20] and have a smaller variance. Additionally, the bitrates of the two cases are similar, indicating that adaptive quantization allows for more fine-grained quantization step adjustments. It does not indiscriminately remove high-frequency information but rather preserves useful high-frequency information. As a result, it can improve reconstruction quality while maintaining the same bitrate. We also present subjective result in the supplementary materials demonstrating the effects of adaptive quantization. The results using adaptive quantization are able to reconstruct more details.

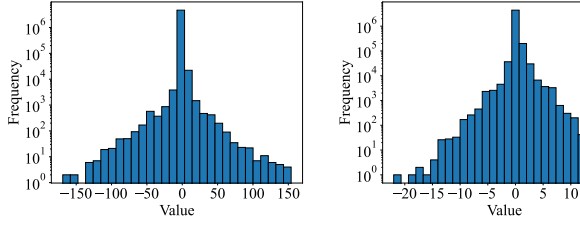

(a) adaptive quantization          (b) w/o adaptive quantization

**Figure 4: Quantized explicit representation amplitude distribution histograms.**

### 4.3 Spatial-Temporal Implicit Entropy Model

Considering that the NeRF representation is learned during training stage, it is plausible to incorporate a rate loss term into the loss function to guide the NeRF representation towards a lower compression bitrate direction. However, it is not feasible to obtain the actual bitrate of the NeRF representation during training stage because entropy coding is non-differentiable. Hence, we propose a spatial-temporal implicit entropy model for accurate rate estimation. Furthermore, unlike the learned entropy model in image compression [33], which is learned from large-scale training datasets, our proposed implicit entropy model is learned on-the-fly with the corresponding NeRF representation in the training stage.

Consequently, for each frame of the NeRF model, we opt to train a specific implicit entropy model, enabling us to accurately estimate the bitrate of the NeRF representation. Furthermore, we encode the implicit entropy model into the bitstream and the decoded entropy model will be used to decode NeRF representation at the decoder side. In particular, we employ the autoregressive entropy model to estimate the bitrate of the explicit NeRF representation. As shown in Fig. 5, the implicit entropy model consists of a two-layer shallow MLP network and incorporates both spatial and temporal context information to estimate the distribution of the NeRF representation.

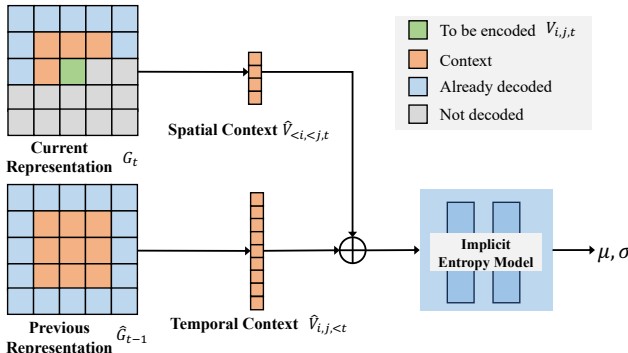

**Figure 5: Illustration of spatial-temporal implicit entropy model. Utilizing the decoded spatial and temporal contexts to predict the distribution of the voxel to be encoded. Although we are actually searching for spatial context in a 3D space, for clarity, we use a 2D plane as an example in the illustration.**

Here, to simplify the description, we assume the $\hat{G}_t$ is 2D representation and the voxel $V_{i,j,t}$ represents the corresponding item in the quantized representation $\hat{G}_t$, i.e., $V_{i,j,t} \in \hat{G}_t$, where $(i,j)$ represents the spatial location. The essence of the autoregressive model is to predict the distribution of undecoded voxel using the already decoded voxel information. As shown in Fig 5, we model the distribution of undecoded voxels using proposed implicit entropy model. Assuming the current voxel to be encoded is $V_{i,j,t}$, we use the spatially adjacent and causally decoded voxels as the spatial context information, denoted as $\hat{V}_{<i,<j,t}$. At the same time, we take the voxel $\hat{V}_{|x-i|\leq 1,|y-j|\leq 1,t-1}$ from the previous frame as the temporal context information, denoted as $\hat{V}_{i,j,<t}$. As illustrated in Fig 5, after concatenating the temporal and spatial context information, we use a super-tiny MLP network to predict the current voxel's probability distribution:

$$\mu, \sigma = f_\psi \left( \hat{V}_{<i,<j,t} \oplus \hat{V}_{i,j,<t} \right). \tag{4}$$

$$p_{\hat{V}_{i,j,t}}(\hat{V}|\hat{V}_{<i,<j,t}, \hat{V}_{i,j,<t}) = \prod_i (\mathcal{N}(\mu,\sigma) * \mathcal{U}(-\tfrac{1}{2}, \tfrac{1}{2}))(\hat{V}_{i,j,t}). \tag{5}$$

where $\mathcal{N}(\mu,\sigma)$ denotes the Laplace distribution.

Then we can calculate the distribution's cumulative function $P_{cdf}(\cdot)$ using the predicted parameters $\mu$ and $\sigma$ and estimate the bitrate of the current voxel:

$$\mathcal{L}_{rate} = \frac{1}{N} \sum_{V_{i,j,t} \in \hat{G}_t} -\log_2 \left[ P_{cdf}\left(\tilde{V}_{i,j,t} + \frac{1}{2}\right) - P_{cdf}\left(\tilde{V}_{i,j,t} - \frac{1}{2}\right) \right]. \tag{6}$$

where $N$ denotes the total number of voxels in the grids, $\tilde{V}_{i,j,t}$ denote the simulated quantized $V_{i,j,t}$.

**Rate-Distortion Loss Function.** Striking a balance between bitrate and distortion is pivotal to efficient NeRF representation compression. Thus, we estimate the bitrate of the NeRF representation and optimize the NeRF representation by using the rate-distortion

loss as follows,

$$\mathcal{L}_{total} = \sum_{\mathbf{r} \in \mathbf{R}} \|C(\mathbf{r}) - \hat{C}(\mathbf{r})\|^2 + \lambda(\mathcal{L}_{rate} + \alpha \mathcal{L}_{reg}). \quad (7)$$

where $C(\mathbf{r})$ is ground truth color, $\hat{C}(\mathbf{r})$ is the rendered color as shown in Equ. 2, and we use the L2 loss as the distortion loss. $\mathcal{L}_{reg}$ essentially serves as a rate constraint. By assigning a larger weight to $\mathcal{L}_{reg}$, the magnitude of the residual grid will be smaller, resulting in a lower bitrate. Therefore, we categorized $\mathcal{L}_{reg}$ and $\mathcal{L}_{rate}$ as bitrate-related loss. $\lambda$ is a trade-off parameter for distortion and rate. To ensure stable training and control the trade-off between quality and bitrate solely through the $\lambda$ parameter, we designed the weights of $\mathcal{L}_{rate}$ and $\mathcal{L}_{reg}$ to have a fixed linear relationship, specifically $\lambda_{reg} = \alpha * \lambda_{rate}$.

### 4.4 Overall Encoding and Decoding Pipeline

In summary, within the NeRF model, the components that need to be compressed and transmitted include the explicit representation, occupancy grid, rendering MLP, and the implicit entropy model. The encoding and decoding of the explicit representation rely on the implicit entropy model, while the other parts can be encoded and decoded independently.

**Occupancy grid.** The characteristic of the occupancy grid is that its elements are either 0 or 1. Thus, for compression, it is first flattened into a vector, grouped in sets of 8 to be packed into uint8 data, and then these uint8 data are compressed. Since a significant portion of the packed uint8 data consists of zeros, the LZMA dictionary encoding algorithm is employed for compression.

**Rendering MLP and Implicit entropy model.** Both the components are MLPs, with the parameters needing compression being the network's weights and biases. Given the small size of these parameters, we simply quantize them and then apply entropy encoding using a range coder [31].

**Explicit representation.** The encoding and decoding of the explicit representation rely on the distribution parameters estimated by the implicit entropy model. Hence, before encoding and decoding the explicit representation, the implicit entropy model must be encoded and decoded first. Based on the distribution parameters estimated by the implicit entropy model, we employ a range coder to encode and decode the explicit representation, following a spatio-temporal causal order.

**Decoding pipeline.** Fig. 6 illustrates the decoding pipeline of our compression framework. Initially, the occupancy grid, rendering MLP, and implicit entropy model are decoded from the bitstream. Subsequently, voxels in the explicit representation are decoded from the bitstream in causal order. Using the already decoded voxels as context, the distribution of undecoded voxels is predicted through an implicit entropy model. Based on the parameters of the distribution, voxels are decoded from the bitstream.

## 5 Experiments

### 5.1 Experimental Setup

**Datasets.** We utilized two datasets: HumanRF [17] and ReRF [48], to validate the effectiveness of our method. For the HumanRF

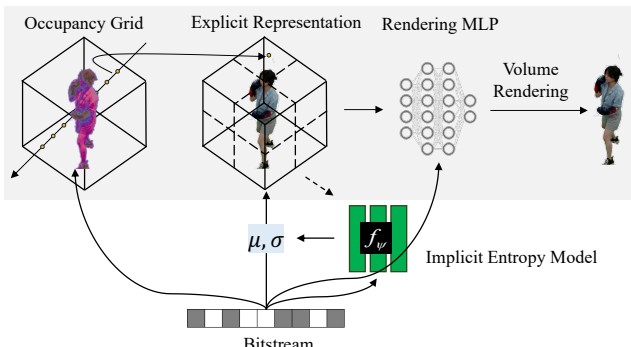

**Figure 6: Decoding pipeline.**

dataset [17], we selected 100 viewpoints with the same camera intrinsics. The image resolution was approximately 1020x750. Among these viewpoints, we designated viewpoints 15 and 25 as the test viewpoints, while the remaining 98 viewpoints were used for training. Regarding the ReRF dataset [17], it comprised approximately 75 viewpoints with an image resolution of 1920x1080. We chose viewpoints 6 and 39 as the test viewpoints following [52], while the rest of the viewpoints were employed for training.

**Evaluation Metrics.** To assess the compression performance of our method on the experimental datasets, we employ Peak Signal-to-Noise Ratio (PSNR), Structural Similarity Index (SSIM) [51], and Learned Perceptual Image Patch Similarity (LPIPS) [55] as quality evaluation metrics. Additionally, we utilize KB per frame as the bitrate evaluation metric. Furthermore, for an overall comparison of compression performance, we utilize the BD-rate metric [2].

**Implementation Details.** In the experiment, we set the group size to 20, which means that an I-frame is inserted every 20 frames within the dynamic scene. For the rate-distortion loss, we defined four different $\lambda$ values($7e-4$, $1e-3$, $2e-3$, and $5e-3$) to achieve different compression ratios. We set $\alpha$ to 10 for the regularization loss. For the training details of NeRF, we refer to the default configuration of DiF [7].

### 5.2 Comparison Results

In our experiments, our primary benchmarks are the state-of-the-art methods TeTriRF [52] and ReRF [48]. Fig. 7 and Fig. 8 illustrate the rate-distortion curves on the HumanRF [17] and ReRF [48] datasets respectively. The figures clearly show that our method achieves the best rate-distortion performance on all metrics—PSNR, SSIM, and LPIPS—both in the training and testing viewpoints. Due to the substantial disparity in the bitrate range between our method and ReRF [48], we only calculate the BD-rate in comparison with TeTriRF [52]. When computing the BD-rate, PSNR is selected as the quality metric. On the HumanRF dataset, our method achieves a BD-rate reduction of **-84.52%** for training viewpoints and **-85.49%** for testing viewpoints. On the ReRF dataset, our method attains a BD-rate reduction of **-60.58%** for training viewpoints and **-76.60%** for testing viewpoints.

**Table 2: The quantitative results on the HumanRF and ReRF datasets. Bold data indicate the best performance, while underlined data indicate the second best.**

| | HumanRF Dataset | | | | | | | ReRF Dataset | | | | | | |
| | training views | | | testing views | | | Size(KB) | training views | | | testing views | | | Size (KB) |
| | PSNR(↑) | SSIM(↑) | LPIPS(↓) | PSNR(↑) | SSIM(↑) | LPIPS(↓) | | PSNR(↑) | SSIM(↑) | LPIPS(↓) | PSNR(↑) | SSIM(↑) | LPIPS(↓) | |
|---|---|---|---|---|---|---|---|---|---|---|---|---|---|---|
| ReRF | 33.60 | 0.960 | 0.111 | 32.27 | 0.950 | 0.128 | 221.86 | 36.08 | 0.972 | 0.046 | 30.69 | 0.961 | 0.055 | 200.25 |
| TeTriRF | 34.84 | 0.965 | 0.096 | 32.49 | 0.954 | 0.115 | 94.14 | 37.65 | 0.977 | 0.039 | 32.21 | 0.968 | 0.045 | 122.15 |
| Ours(Low) | 36.50 | 0.969 | 0.090 | 33.87 | 0.961 | 0.107 | **55.72** | 38.41 | 0.980 | 0.035 | 32.47 | 0.970 | 0.043 | **67.02** |
| Ours(High) | **37.40** | **0.973** | **0.079** | **34.30** | **0.965** | **0.095** | 80.63 | **39.28** | **0.984** | **0.031** | **32.70** | **0.973** | **0.039** | 116.64 |

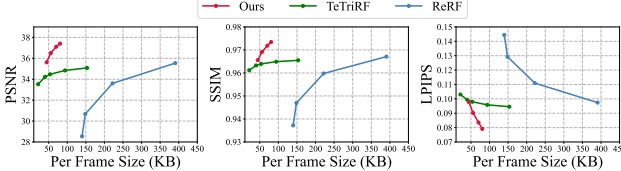

**(a) Rate-distortion performance for the training views.**

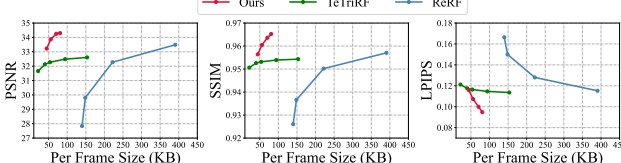

**(b) Rate-distortion performance for the testing views.**

**Figure 7: The rate-distortion performance comparison results on HumanRF dataset. The BD-rate of our method compared to TeTriRF is as follows: in the training viewpoints, the BD-rate is -84.52%, and in the testing viewpoints, the BD-rate is -85.49%.**

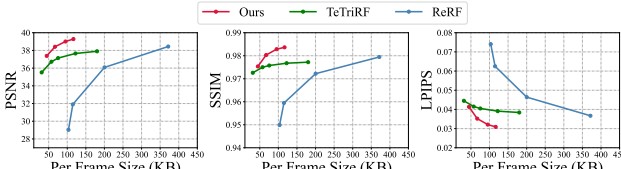

**(a) Rate-distortion performance for the training views.**

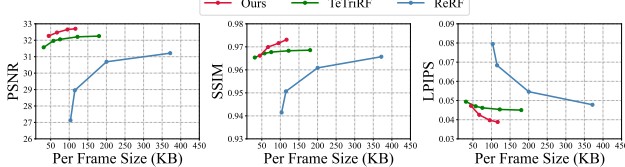

**(b) Rate-distortion performance for the testing views.**

**Figure 8: The rate-distortion performance comparison results on ReRF dataset. The BD-rate of our method compared to TeTriRF is as follows: in the training viewpoints, the BD-rate is -60.58%, and in the testing viewpoints, the BD-rate is -76.60%.**

**Table 3: The comparison results of our method applied on TensoRF [6] and the baseline applied on TensoRF.**

| Ours (TensoRF) | | | | Baseline (TensoRF) | | | |
|---|---|---|---|---|---|---|---|
| PSNR | SSIM | LPIPS | Size (KB) | PSNR | SSIM | LPIPS | Size(KB) |
| 33.39 | 0.966 | 0.093 | 58.07 | 33.65 | 0.970 | 0.081 | 1745.43 |
| 32.30 | 0.963 | 0.099 | 47.14 | 32.45 | 0.962 | 0.097 | 1254.91 |
| 31.02 | 0.954 | 0.116 | 39.59 | 30.84 | 0.948 | 0.117 | 999.48 |
| 29.24 | 0.943 | 0.132 | 30.82 | 27.03 | 0.905 | 0.154 | 671.51 |

Tab. 2 shows the detailed quantitative results on HumanRF dataset and ReRF dataset, where Ours(Low) and Ours(High) respectively represent the results of our method under different $\lambda$ configurations. The table indicates that both our method and TeTriRF can achieve good reconstruction quality at lower bitrate range, while ReRF requires significantly higher bitrate for similar quality level. Moreover, Ours(Low) achieves a bitrate substantially lower than TeTriRF with comparable reconstruction quality. And Ours(High) offers much higher reconstruction quality at a bitrate similar to TeTriRF, further demonstrating the rate-distortion performance advantage of our method.

Fig. 9 displays qualitative comparison results on the HumanRF's *actor07* sequence and the ReRF's *kpop* sequence. The figure illustrates that our method can reconstruct more details at a lower bitrate, such as the beard in *actor07* and the clothing logo in *kpop*, which are lost in the reconstructions by TeTriRF and ReRF. This demonstrates that our method also provides a superior subjective experience.

## 5.3 Evaluation on Plane Representaion

Our method is universally applicable to a large amount of explicit representation. To verify the effectiveness of our method, we further evaluated its performance on the plane-based representation TensoRF [6]. The comparison baseline is: modeling each frame in dynamic scenes with an independent TensoRF model, followed by quantization and entropy encoding of TensoRF after modeling is complete.

Tab. 3 shows the comparison results between our method and the baseline method at various compression ratios. It is evident that our method achieves significantly lower bitrates than the baseline method, without a noticeable decline in reconstruction quality. This demonstrates that our method is not only suitable for DiF [7] but also applicable to TensoRF [6], proving its effectiveness across different explicit representations.

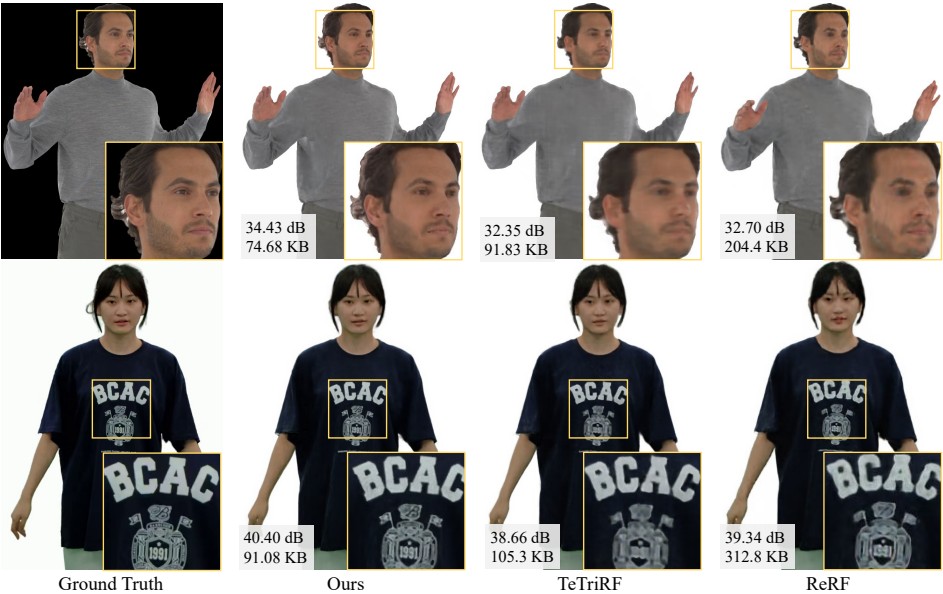

| 34.43 dB | 32.35 dB | 32.70 dB |
|----------|----------|----------|
| 74.68 KB | 91.83 KB | 204.4 KB |

| 40.40 dB | 38.66 dB | 39.34 dB |
|----------|----------|----------|
| 91.08 KB | 105.3 KB | 312.8 KB |

Ground Truth      Ours      TeTriRF      ReRF

**Figure 9: Qualitative comparisons of *actor07* sequence from HumanRF [17] and *kpop* sequence from ReRF [48]. PSNR and size results are given at lower-left.**

## 5.4 Ablation Studies

In the ablation studies, we aim to verify the efficacy of inter prediction-based dynamic modeling, adaptive quantization, and rate-distortion joint optimization we proposed. We conduct ablation studies on the *actor02* sequence from the HumanRF [17] dataset. Based on the full method, we disable dynamic modeling and adaptive quantization separately during training. Tab. 4 shows the bitrates for different ablation experiments at the same quality level (PSNR = 34 dB). The definition of "Baseline" is similar to Sec. 5.3, where each frame of the dynamic scene is modeled using an independent DiF model, followed by quantization and entropy encoding of the representation after modeling is complete.

**Table 4: At the same quality level (PSNR = 34 dB), the bitrates for different ablation experiments. "w/o Dynamic" signifies the disabling of dynamic modeling, and "w/o Adaptive" indicates the disabling of adaptive quantization.**

| Baseline | w/o Dynamic | w/o Adaptive | Full |
|----------|-------------|--------------|------|
| 1193.29 KB | 137.45 KB | 80.31 KB | 66.61 KB |

From Tab. 4, it is apparent that, based on the "Full" method, either disabling dynamic modeling or adaptive quantization results in an increase in bitrate, indicating the effectiveness of these two modules. Comparing the "Baseline" with the other three experiments, it is evident that the bitrates of the other three experiments are significantly lower than that of the "Baseline", demonstrating the efficacy of our proposed rate-distortion joint optimization.

Moreover, we analyze the average bit consumption for I-frame and P-frame under different $\lambda$ configurations, and the result is illustrated in Tab. 5. It is evident that the bit consumption of P-frames is significantly lower than that of I-frames, which validates

**Table 5: Analysis of average bit consumption for I-frame and P-frame.**

|  | $\lambda = 0.0007$ | $\lambda = 0.001$ | $\lambda = 0.002$ | $\lambda = 0.005$ |
|---|---|---|---|---|
| I-frame (KB) | 272.28 | 207.69 | 161.42 | 116.25 |
| P-frame (KB) | 94.05 | 79.67 | 61.28 | 46.48 |

the effectiveness of our dynamic modeling method in removing inter-frame redundancy and reducing the bitrate of P-frames.

## 6 Conclusion

In this paper, we present a dynamic modeling and rate-distortion optimization framework tailored for explicit NeRF representation. By incorporating a rate-aware training strategy into the NeRF training process, our approach demonstrates enhanced compression efficiency. Utilizing an implicit entropy model for accurate rate estimation of NeRF representations, coupled with adaptive quantization, we further amplify the compression efficacy. Experimental results reveal that our method secures the highest compression efficiency across two benchmark datasets. The NeRF-based modeling and compression methodology we propose is capable of reducing the data volume of a single frame of radiance field to below 100KB, markedly advancing the transmission of volumetric videos.

## Acknowledgments

This work was supported by National Key RD Project of China (2019YFB1802701), National Natural Science Foundation of China (62102024, 62331014), the Fundamental Research Funds for the Central Universities; in part by the 111 project under Grant B07022 and Sheitc No.150633; in part by the Shanghai Key Laboratory of Digital Media Processing and Transmissions.

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
