# OpenReview forum: "Rate-aware Compression for NeRF-based Volumetric Video"
_acmmm.org/ACMMM/2024/Conference — MM2024 Oral_

### Official Review · Reviewer_RAos · 2024-05-09

**Rating:** 4
**Confidence:** 2

**Summary:**

Inspired by traditional compression approaches, the authors creatively introduced the spatial-temporal implicit entropy model to reduce redundant information. An adaptive quantization strategy further enhances the performance of the framework. Additionally, due to the well-designed framework, it can be used for other explicit NeRF-based models.

**Strengths:**

1. Authors have successfully identified and distilled the key ideas of the paper and systematically validated them through experiments. In particular, categorizing the main ideas has been very effective and added clarity to the overall paper.
2. The method is novel and experiments demonstrate that the proposed framework achieves superior performance while also exhibiting a degree of applicability to other explicit NeRF-based models.

**Limitations:**

1. Missing ", "or ". "at the end of all formulas in the paper.
2. The description of Figure 4 is not clear enough.
3. It is regrettable that validation has been only limited to performance aspects. In this regard, more diverse experimental validations are needed, such as comparing training time with baseline.
4. It is regrettable that the framework has not been validated on other types of datasets like DyNeRF.

**Suitability:**

2

---

### Official Review · Reviewer_LExs · 2024-05-18

**Rating:** 5
**Confidence:** 4

**Summary:**

In this paper, the authors propose a rate-aware compression framework for NeRF-based volumetric video compression, and propose to introduce the entropy model module and adaptive quantization strategy to achieve higher compression efficiency. The proposed method can be used for compression of different representations, and achieves state-of-the-art rate-distortion performance for volumetric video.

**Strengths:**

++ Clear Motivation. The entropy coding and adaptive quantization are introduced into volumetric video compression task, it’s a milestone work.

++ Good Performance. The proposed method achieves the SOTA performance, and significantly improves the compression efficiency.

++ Well Written. The paper is easy to understand.

++ Well Design. The effectiveness of proposed modules is verified in ablation experiments.

**Limitations:**

-- Some Advices.

1) In this paper, author follows the reference structure of I-P-P, how about the reference structure I-B-B in this task? I think the efficient reference structure can further improve this work.

2) In the design of spatial-temporal implicit entropy model, the autoregressive entropy model is used, how about the cooperation of hyperprior and autoregressive entropy model in this work? In learned video compression, many works are follow this manner [1].

Ref: [1] Li J, Li B, Lu Y. Deep contextual video compression [J]. Advances in Neural Information Processing Systems, 2021, 34: 18114-18125.

-- Lack of Reference.

1) Some volumetric image\video compression references are missing.

[1] Xue D, Ma H, Li L, et al. aiWave: Volumetric image compression with 3-D trained affine wavelet-like transform [J]. IEEE Transactions on Medical Imaging, 2022, 42(3): 606-618.

2) Some milestone end-to-end video compression references are missing in related work.

[1] Li J, Li B, Lu Y. Neural video compression with diverse contexts[C]. Proceedings of the IEEE/CVF Conference on Computer Vision and Pattern Recognition. 2023: 22616-22626.

[2] Sheng X, Li L, Liu D, et al. VNVC: A Versatile Neural Video Coding Framework for Efficient Human-Machine Vision[J]. IEEE Transactions on Pattern Analysis and Machine Intelligence, 2024.

**Suitability:**

3

---

### Official Review · Reviewer_JZqn · 2024-05-25

**Rating:** 6
**Confidence:** 3

**Summary:**

This paper proposes a compression model for NeRF representation, which combines the compression of NeRF and the NeRF training together. Experimental results verify the effectiveness of the proposed method, in terms of various quality metrics. This paper is a pioneer work considering the NeRF data compression in training phase, which is an interesting idea.

**Strengths:**

Compared with existing NeRF compression methods, the proposed method could achieve significant performance improvement, indicating the efficiency of the proposed method.

There are some highlights for the proposed NeRF compression model:
1. The temporal correlation is well modeled with a inter prediction based dynamic modeling
2. Adaptive quantization, which could bear more high-frequency information under the similar bitrate, compared with fixed quantization.
3. A joint loss function designed for NeRF compact representation.

**Limitations:**

The complexity of the proposed algorithm should be investigated, and compare the complexity of the proposed method with other popular methods.

generally speaking, in signal compression models, the loss function are usually formulated with two parts, the bitrate one and the distortion one. However, regarding the compression distortions, the color distortion is an individual part in loss function. But, the grid distortion is combined with the bitrate component. Please give some explanation about the reason of this formulation.

In equation 7, the \alpha is fixed 10, and \lambda is varied to achieved various compression ratios. Why \alpha is fixed?

**Suitability:**

3

---

### Meta-Review · Area_Chair_gNZr · 2024-07-04

**Recommendation:** Accept (Oral)
**Confidence:** 4

**Metareview:**

The paper proposes a rate-aware compression model for Nerfs.  The proposed approach includes a rate objective during the training stage, which allows it to be optimized using a rate-distortion trade-off.  The reviewers agree that the paper should be accepted.  Among other things, the reviewers mention that the motivation is clear, the proposed method is novel, the paper is well-written, and it performs well.  In the rebuttal, the authors also answered the main concerns raised by the reviewers, including an analysis of the complexity of the model, a better explanation of the loss function, and a supplement experiment on the DyNeRF dataset.

 After a thorough evaluation, including the reviewers' feedback and the authors' rebuttal, I recommend the paper be accepted.